# Six Years of Grassland Cultivation Promotes CO₂, N₂O Emissions and CH₄ Uptake with Increasing N Deposition on Qinghai-Tibetan Plateau

**Hang Shi [1], Hao Shen [1,2,*], Shikui Dong [1,2,3,*], Jiannan Xiao [2], Zhiyuan Mu [2], Ran Zhang [1], Xinghai Hao [1], Ziying Wang [1] and Hui Zuo [1]**

[1] School of Grassland Science, Beijing Forestry University, Beijing 100083, China
[2] State Key Joint Laboratory of Environmental Simulation and Pollution Control, School of Environment, Beijing Normal University, Beijing 100875, China
[3] Department of Natural Resources, Cornell University, Ithaca, NY 14853, USA
* Correspondence: shenhao2222@gmail.com (H.S.); dongshikui@bjfu.edu.cn (S.D.)

**Abstract:** Nitrogen (N) deposition has become an important factor of vital changes in the Qinghai-Tibetan Plateau (QTP), one of the key eco-regions in the world. To investigate how N deposition affects the fluxes of GHGs (CH₄, CO₂, N₂O) in the alpine grassland ecosystem, the dominant ecosystems on QTP, we conducted control experiments in three types of alpine grasslands, including the alpine meadow (AM), alpine steppe (AS), and cultivated grassland (CG) on the QTP. In this study, four N addition gradients (0 kg Nha⁻¹yr⁻¹, 8 kg Nha⁻¹yr⁻¹, 24 kg Nha⁻¹yr⁻¹, and 40 kg Nha⁻¹yr⁻¹) were set up using ammonium nitrate from 2015 to 2020 in order to simulate N deposition at different levels, and after 6 years of continuous N application, greenhouse gases were collected from sampling plots. The results showed that simulated N deposition had no significant effect on soil GHG fluxes, while the grassland type had an extremely significant effect on soil GHG fluxes. Under the same N deposition conditions, the CH₄ absorption in the cultivated grassland was higher than that in the other two types of grasslands. At low N deposition levels (CK, N1), the CO₂ emission in the cultivated grassland was higher than that in the other two types of grasslands. At high N deposition levels (N2 and N3), the N₂O emission in the cultivated grassland increased more significantly than it did in the other two types of grasslands. Control of grassland cultivation should be proposed as a reliable form of land-use management to reduce GHG emissions on the QTP in the era of increasing N deposition.

**Keywords:** nitrogen deposition; greenhouse gas; Qinghai-Tibetan Plateau; grassland type; soil factor

## 1. Introduction

The continuous increase of greenhouse gas (GHGs) emissions, particularly those from CH₄, CO₂, and N₂O [1], has led to further increases in global temperatures, which have a huge impact on the global economy and ecosystems [2]. As an important component of the Earth system, grassland ecosystems play a crucial role in the global carbon and N cycles. The response of greenhouse gas fluxes to atmospheric N deposition might be different in various grassland ecosystems [3–7].

Increasing atmospheric N deposition has become a serious global issue [8]. The rate of N input into terrestrial ecosystems continues to increase, and the use of fossil fuels, N fertilizers as well as accelerated urbanization have all contributed to the accumulation of atmospheric N, followed by atmospheric N landing as N-containing particles, and soluble N [9]. Global N deposition is predicted to increase by 2.5 times in the future [10]. In the past 30 years, with the gradual improvement of China's industrialization level and the intensification of agricultural activities, China has become one of the top three countries

in the world to experience continuously increased N deposition [11–14], with an average rate of 8 kg ha$^{-1}$ of atmospheric N deposition per year. The continuous input of N into the ecosystems will affect the carbon and N cycles and eventually lead to system disruption [15]. The series of N reactions and the cycling process in plants are the main pathways of N movement in grassland ecosystems, and therefore grassland ecosystems must change with rising N content in the soil [16]. The increase of N deposition can significantly inhibit the $CH_4$ uptake and increase the $N_2O$ emissions of soils with various land-use patterns, while it has a non-significant positive effect on soil $CO_2$ emissions [7]. The effects of N deposition on GHGs vary with the types of grasslands, e.g., some scholars noted that the accumulation of N deposition tends to inhibit $CH_4$ uptake in alpine grasslands [3] and temperate grasslands [6], while other scholars found that N deposition significantly increases $CH_4$ uptake [4] or has no significant effect on changes in $CH_4$ fluxes in alpine grasslands [17]. In terms of $CO_2$ fluxes, Jiang et al. (2010) [3] found that N deposition had a tendency to insignificantly reduce the $CO_2$ emissions of alpine meadows throughout the growing season. Some researchers have drawn the opposite conclusion, finding that the accumulation of N deposition significantly increased $CO_2$ emissions [4] or did not alter $CO_2$ fluxes in temperate grasslands [6] and alpine steppes [17]. For temperate grasslands, Chen et al. (2019) found that increased N deposition promoted $N_2O$ emissions [6], which is consistent with the conclusions reached by other scholars on alpine grasslands [4,13]. The above results suggest that the response of soil GHGs to N deposition is not constant, and that the intrinsic response mechanisms are complex [7,18,19].

With an average altitude of over 4000 m, the Qinghai-Tibetan Plateau (QTP) is known as the "Water Tower of Asia" [20] and is an important ecological barrier in China [20]. The QTP's grassland ecosystems, which are the dominant ecosystems of the plateau, have an important implication in the study of global ecological change as they are not only sensitive to changes in the external environment but are also influenced by increased atmospheric N deposition [21]. Previous studies have shown that N deposition promotes $CO_2$ emissions from alpine meadows on the QTP [22–24]. Meanwhile, it has also been suggested that $CO_2$ emissions from alpine meadows on the QTP decreased under N deposition conditions [3,18]. There have been conflicting results on the effect of N application on $N_2O$ emissions from alpine grassland ecosystems [3]. Therefore, in this study, we investigated how N deposition affected GHG fluxes in the alpine meadow (AM), alpine steppe (AS), and cultivated grassland (CG) during the growing season, which was based on the hypothesis that simulated N deposition could change GHG emissions in the alpine grasslands, and that the grassland types can mediate the effects of N deposition on GHG fluxes in alpine regions.

## 2. Materials and Methods

### 2.1. Study Sites

This study was conducted in Xihai Town, Haiyan County, Qinghai Province, China (100°95′ E, 36°93′ N) and in Tiebujia Town, Gonghe County, Qinghai Province, China (99°35′ E, 37°05′ N). The average elevation of Xihai Town is 3100 m, while the average annual temperature, the mean annual precipitation, and the annual evaporation capacity are 1.4 °C, 330–370 mm, and 1400 mm, respectively; the grassland type in Xihai Town is a typical alpine meadow with clay-based soil. The dominant species is *Kobresia capillifolia* (Decne.) C. B. Clarke. The average elevation of Tiebujia Town is 3200 m, while the average annual temperature, the mean annual precipitation, and the annual evaporation capacity are 0 °C, 360–430 mm, and 1550 mm, respectively; the grassland type in Tiebujia Town is alpine steppe, and the dominant species is *Stipa capillata* L. The cultivated grassland site was also located in Tiebujia Town, next to the alpine steppe; *Elymus nutans* was planted in the cultivated grassland in 2012.

### 2.2. Experimental Design

A randomized block design was adopted in this experiment. A total of 12 rectangular plots of 2 m by 5 m were selected in each of the three grassland types, namely the alpine meadow (AM), alpine steppe (AS), and cultivated grassland (CG), with a 1 m buffer zone between the plots. Different N deposition gradients were set, and ammonium nitrate ($NH_4NO_3$, Nitrogen content: 35%) was used to fertilize. Four N addition rates—CK (0 kg $Nha^{-1}yr^{-1}$), N1 (8 kg $Nha^{-1}yr^{-1}$), N2 (24 kg $Nha^{-1}yr^{-1}$), and N3 (40 kg $Nha^{-1}yr^{-1}$)—were set to simulate different levels of N deposition based on the average N deposition of 8 kg $Nha^{-1}yr^{-1}$ [25] and 50 kg $Nha^{-1}yr^{-1}$ critical loads on the QTP [26]. There were three replication plots randomly selected for each N application treatment. All treatments were based on the same land-use history and similar topography. From 2015 to 2020, we applied fertilizer in early May and early July each year to simulate N deposition.

### 2.3. Greenhouse Gases—Sampling and Measurements

After six years of treatment, greenhouse gases were collected in early August, the peak of the 2020 growth season. Sampling was conducted on three consecutive sunny days, from 9 am to 11 am. An opaque PVC static box with a diameter of 20 cm and a height of 25 cm was used for sampling (Figure 1); the box was placed on a base (the base was made of PVC and had been inserted 5 cm deep into the soil) and sealed with distilled water. The gas inside the chamber was collected with the use of 50 mL plastic syringes 0, 5, 10, 15, 20, and 25 min after closing the static chamber [27]. The collected gas samples were sealed in foil sampling bags and then transported to the laboratory within 24 h; the samples were analyzed using a gas chromatograph (Agilent-7890B, Agilent Technologies, Inc. Santa Clara, CA, USA).

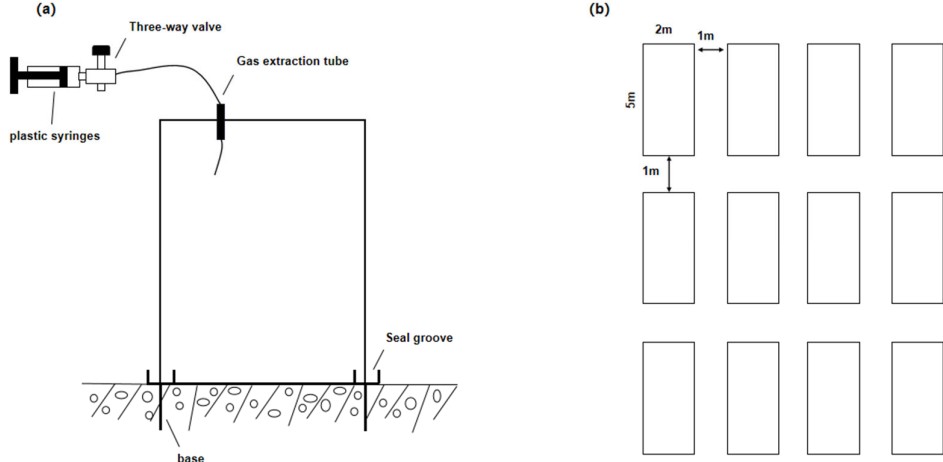

**Figure 1.** (**a**) Sampling device; (**b**) Plot setting diagram.

### 2.4. Soil Sampling

Together with GHG sampling (early August), we sampled the soil in each plot with the use of a soil auger that was 4 cm in diameter. After the soil was dried naturally and sieved using a 0.25 mm mesh, we sealed the soil samples in polyethylene bags and transported them to the laboratory for extraction. Soil total N (TN) and total carbon (TC) were measured using an element analyzer (EuroVector-EA 3000, Pavia, Italy). Soil AP, Ca, K, P, and S were measured using inductively coupled plasma spectrometers (ICP) (SPECTRO ARCOSEOP, Kleve, Germany) [5]. Soil pH was determined from the filtered supernatant using a pH meter (Mettler Toledo, Zurich, Switzerland). Soil $NH_4^+$-N and $NO_3^-$-N content were measured using a flow injection auto-analyzer (Systea, Rome, Italy) [5].

### 2.5. Statistical Analysis

Two-way analysis of variance (ANOVA) was used in SPSS 23.0 (SPSS Inc. Chicago, IL, USA) to examine the differences in the GHG fluxes' responses to different N application treatments in the different grassland types. Duncan's Multiple Range Test (Duncan) in the analysis of variance (ANOVA) was used to test the differences in the GHG response of different grassland types. Pearson correlation analyses were used to examine the relationships between the average greenhouse gas fluxes and the soil properties after the different treatments.

## 3. Results

### 3.1. N Deposition Effects on $CH_4$ Fluxes in Different Types of Grasslands

As can be seen in Table 1, the effect of grassland type on the $CH_4$ flux was extremely significant. Overall, the effect of N deposition level on $CH_4$ emission was fairly weak as compared to that of grassland type; insignificant interactions between grassland type and N deposition level were found (Table 1). $CH_4$ emission in the alpine meadow and the cultivated grassland was less affected by N deposition level, while low N deposition significantly restrained $CH_4$ uptake in the alpine steppe (Figure 2a). At the same N deposition levels, the cultivated grassland presented a much higher $CH_4$ uptake than the other two types of natural grasslands (Figure 2b).

**Table 1.** Effects of grassland type (GT), N deposition level (NDL), and their interactions (GT×NDL) on GHG fluxes.

|  | GT (F-Value) | NDL (F-Value) | GT×NDL (F-Value) |
| --- | --- | --- | --- |
| $CH_4$ | 19.886 *** | 1.685 | 0.509 |
| $CO_2$ | 8.866 *** | 2.425 | 1.403 |
| $N_2O$ | 16.273 *** | 0.829 | 0.822 |

Note: *** indicates $p < 0.001$.

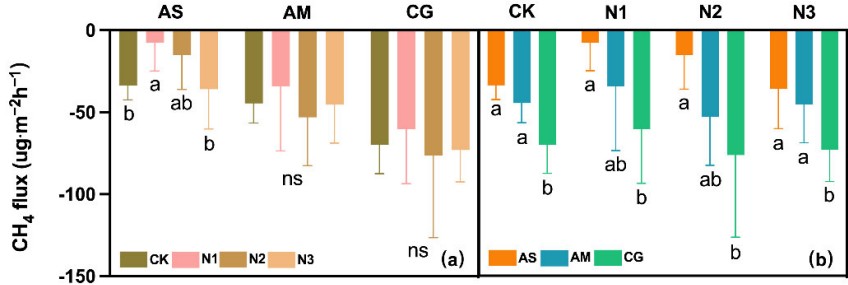

**Figure 2.** (**a**) The impact of N deposition levels on $CH_4$ fluxes; (**b**) The impact of grassland type on $CH_4$ fluxes. Different letters indicate a significant difference between different grassland types at $p < 0.05$. AM: alpine meadow; AS: alpine steppe; CG: cultivated grassland; CK: no N deposition; N1: N addition at 8 kg Nha$^{-1}$yr$^{-1}$; N2: N addition at 24 kg Nha$^{-1}$yr$^{-1}$; N3: N addition at 40 kg Nha$^{-1}$yr$^{-1}$.

### 3.2. Effects of N Deposition on $CO_2$ Fluxes in Different Grasslands

In this study, $CO_2$ emissions were significantly affected by grassland type (Table 1). The effect of N deposition level on $CO_2$ emission was fairly weak as compared to that of grassland type; no obvious interactions between grassland type and N deposition level were found (Table 1). The N deposition level had an insignificant effect on $CO_2$ emission in both the alpine steppe and the cultivated grassland, while high N deposition levels significantly increased $CO_2$ emissions in the alpine meadow (Figure 3a). In addition, we found that the cultivated grassland showed significantly higher $CO_2$ emissions than two types of natural grasslands at low N deposition levels (Figure 3b).

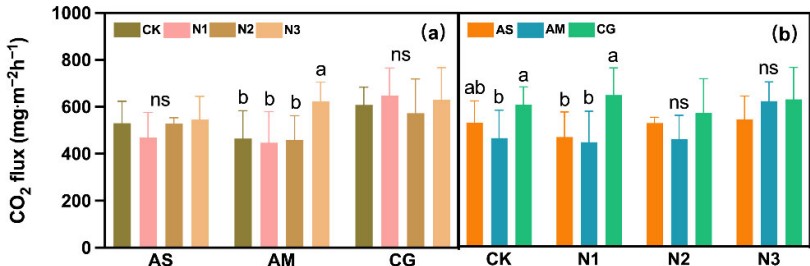

**Figure 3.** (**a**) The impact of N deposition level on $CO_2$ fluxes. (**b**) The impact grassland type on $CO_2$ fluxes. Different letters indicate significant differences between different grassland types at $p < 0.05$. AM: alpine meadow; AS: alpine steppe; CG: cultivated grassland; CK: no N deposition; N1: N addition at 8 kg Nha$^{-1}$yr$^{-1}$; N2: N addition at 24 kg Nha$^{-1}$yr$^{-1}$; N3: N addition at 40 kg Nha$^{-1}$yr$^{-1}$.

### 3.3. Effects of N Deposition on N₂O Fluxes in Different Grasslands

$N_2O$ emission was significantly affected by grassland type (Table 1). The N deposition level and its interaction with grassland types insignificantly affected $N_2O$ emission (Table 1). The N deposition level had insignificant effects on soil $N_2O$ emissions in the alpine meadow and the cultivated grassland, while high N deposition significantly enhanced $N_2O$ emissions in the alpine steppe (Figure 4a). In this study, we found that the cultivated grassland produced higher $N_2O$ emissions than two types of native grasslands at high N deposition levels (Figure 4b).

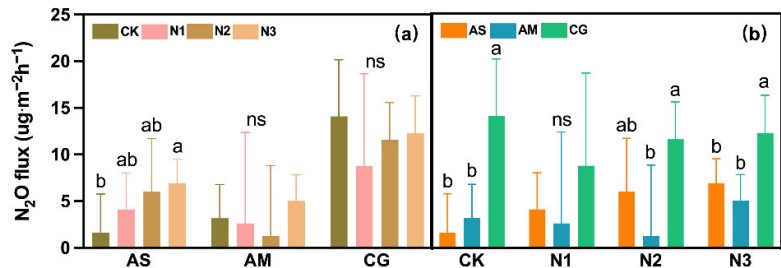

**Figure 4.** (**a**) The impact of N deposition level on $N_2O$ fluxes. (**b**) The impact of grassland type on $N_2O$ fluxes. Different letters indicate significant differences between different grassland types at $p < 0.05$. AM: alpine meadow; AS: alpine steppe; CG: cultivated grassland; CK: no N deposition; N1: N addition at 8 kg Nha$^{-1}$yr$^{-1}$; N2: N addition at 24 kg Nha$^{-1}$yr$^{-1}$; N3: N addition at 40 kg Nha$^{-1}$yr$^{-1}$.

### 3.4. Relations between Soil Properties and GHG Emissions in Different Grasslands

In the alpine steppe (AS), the $CO_2$ emissions showed a significant positive correlation with soil TN, $CH_4$ showed a significantly negative correlation with soil P, and the $N_2O$ emissions were significantly negatively correlated with AP (Figure 5a and Table 2). In the alpine meadow (AM), we found that only soil $NO_3^-$-N showed a significant positive correlation with the $N_2O$ emissions (Figure 5b and Table 2). In the cultivated grasslands (CG), we found that soil P showed a significantly positive correlation with the $CO_2$ emissions, and that soil K showed a significantly negative correlation with the $N_2O$ emissions (Figure 5c and Table 2).

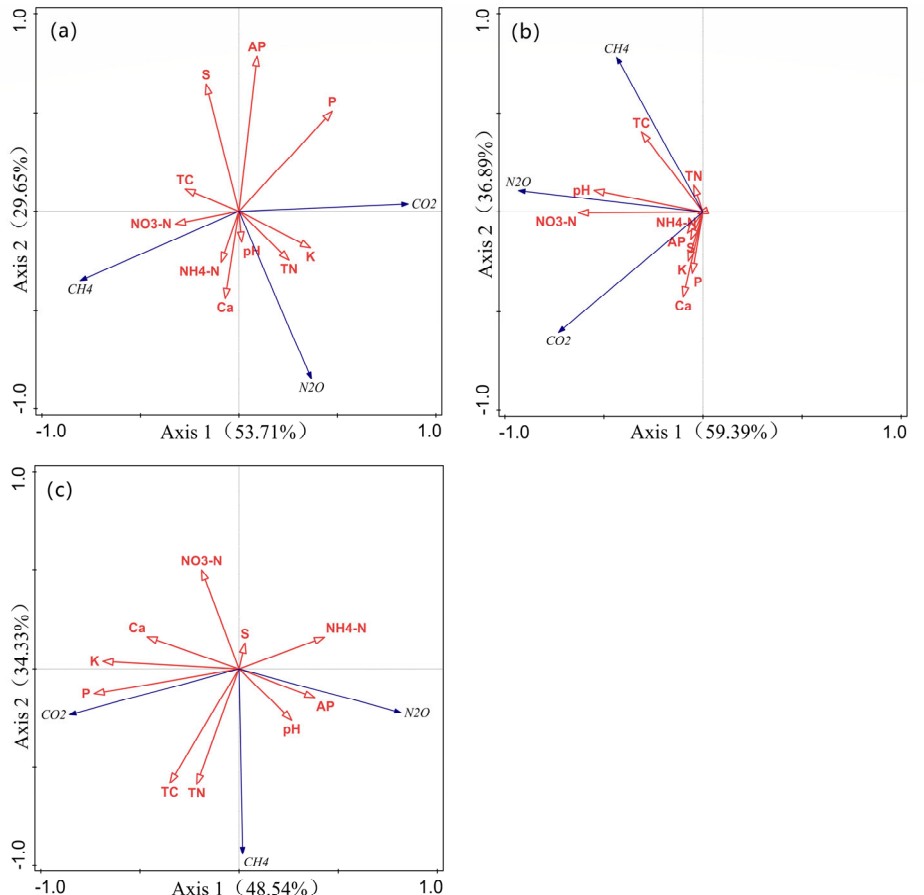

**Figure 5.** RDA (redundancy analysis) between GHG fluxes and soil factors in different types of grasslands. (**a**) Alpine steppe (AS); (**b**) Alpine meadow (AM); (**c**) Cultivated grassland (CG). Soil factors: TN (total nitrogen); TC (total carbon); $NH_4^+$-N (ammonium nitrogen); $NO_3^-$-N (nitrate nitrogen); AP (available phosphorus); Ca (Calcium); K (potassium); P (Phosphorus); S (Sulfur).

**Table 2.** Person's correlation test between GHG fluxes and soil nutrients.

| Soil | AS | | | AM | | | CG | | |
|---|---|---|---|---|---|---|---|---|---|
| Factors | $CH_4$ | $CO_2$ | $N_2O$ | $CH_4$ | $CO_2$ | $N_2O$ | $CH_4$ | $CO_2$ | $N_2O$ |
| TN | 0.238 | 0.579 * | 0.178 | 0.186 | 0.018 | −0.023 | 0.542 | 0.336 | −0.026 |
| TC | 0.132 | −0.286 | −0.178 | 0.478 | 0.009 | 0.301 | 0.52 | 0.478 | −0.11 |
| $NH_4^+$-N | 0.337 | 0.103 | 0.121 | 0.003 | 0.023 | 0.024 | −0.123 | −0.443 | 0.278 |
| $NO_3^-$-N | 0.066 | −0.516 | 0.014 | 0.268 | 0.458 | 0.591 * | −0.343 | −0.211 | −0.538 |
| AP | −0.28 | 0.184 | −0.662 * | −0.055 | 0.114 | 0.071 | 0.23 | −0.455 | 0.174 |
| Ca | 0.261 | −0.019 | 0.328 | −0.301 | 0.32 | 0.062 | −0.149 | 0.338 | −0.44 |
| K | −0.35 | 0.173 | 0.334 | −0.165 | 0.198 | 0.049 | −0.044 | 0.57 | −0.580 * |
| P | −0.614 * | 0.367 | −0.237 | −0.217 | 0.225 | 0.018 | 0.088 | 0.695 * | −0.529 |
| S | −0.222 | −0.263 | −0.561 | −0.084 | 0.124 | 0.048 | −0.176 | 0.051 | 0.109 |
| pH | 0.29 | 0.276 | 0.048 | 0.341 | 0.354 | 0.503 | 0.291 | −0.238 | 0.2 |

Note: AS, alpine steppe; AM, alpine meadow; CG, cultivated grassland; TN, total nitrogen; TC, total carbon; $NH_4^+$-N, ammonium nitrogen; $NO_3^-$-N, nitrate nitrogen; AP, available phosphorus; Ca, Calcium; K, potassium, P, Phosphorus; S, Sulfur. * Indicates the significance level $p < 0.05$.

## 4. Discussion

### 4.1. Effects of N Deposition on Soil GHG Emissions in Alpine Grasslands

The absorption capacity of $CH_4$ for long-wave radiation is 20–30 times greater than that of the equivalent $CO_2$, which makes $CH_4$ the second most important greenhouse gas after $CO_2$, contributing about 18% to global warming [28]. $CH_4$ uptake by grassland ecosystems is mainly influenced by soil water content, soil temperature and humidity, and microbial activity [29]. The results of the present study showed that $CH_4$ fluxes were negative in all alpine grassland ecosystems on the Qinghai-Tibetan Plateau during the growing season, suggesting that alpine grassland ecosystems may be acting as a sink for $CH_4$ (Figure 2a). This has been confirmed in previous studies [18]. In this study, we found that N deposition did not affect $CH_4$ uptake in the AM and CG, but it inhibited $CH_4$ uptake in the AS, particularly at a low N addition level. This is consistent with the results obtained from an alpine meadow by Jiang et al. (2010) [3]. However, it is inconsistent with the results obtained by Zhang et al. (2013) [18], who found that low N treatment promoted $CH_4$ uptake in an alpine meadow of the Eastern Qinghai-Tibetan Plateau, while medium and high N treatments inhibited soil $CH_4$ uptake. Such phenomena can be explained by the fact that the N addition may inhibit both $CH_4$ production and $CH_4$ oxidation, depending on the relative magnitude of inhibition in both processes [30]. It may also be that the nitrogen fertilizer used in their study was $NH_4Cl$, $(NH_4)_2SO_4$, $KNO_3$, which is different from that used in our study.

In grassland ecosystems, $CO_2$ emissions are mainly derived from ecosystem respiration, which includes autotrophic and heterotrophic respiration. In a study conducted by Bowden et al. (2004) [31], it was noted that N application had no significant effect on heterotrophic respiration but had a significant inhibitory effect on autotrophic respiration. Heterotrophic respiration is mainly affected by environmental conditions such as soil nutrient conditions and soil microbial communities, which can be altered by N application [30]. In this study, $CO_2$ emissions were positive in all three different types of grasslands on the QTP throughout the growing season, indicating that alpine grassland ecosystems are a source of $CO_2$. In addition, the results of this study showed that high levels of N application significantly promoted $CO_2$ emissions in the alpine meadow (AM) on the Qinghai-Tibetan Plateau. This is consistent with the results of Zong et al. (2013) and Li et al. (2014) [23,24]. The possible causes of increased $CO_2$ emissions in the grassland ecosystems with N deposition are increased microbial activity [24], increased biomass [23,24], and increased soil organic N content [23,24].

$N_2O$ emissions in grassland ecosystems depend mainly on soil microbial nitrification and denitrification [32]. However, this process is strongly influenced by environmental factors [33,34]. In the present study, we found that $N_2O$ emissions were significantly increased at high N levels in the alpine steppe. This is consistent with the results obtained by previous researchers from alpine meadows, temperate grasslands, and grasslands [3,13,35]. In contrast, some other researchers pointed out that $N_2O$ fluxes were not altered by N addition [17]. This can be explained by the fact that different $N_2O$ emissions may be associated with diverse factors and processes such as climate, soil texture, soil organic carbon, soil microbial nitrification and denitrification processes as well as cation exchange [17,36]. In contrast to this study, Wei et al.'s research site was 4730 m above sea level, and the annual precipitation was about 414.6 mm [17], which may be one of the reasons for the different results.

### 4.2. Grassland Types Matter in Mediating Effects of N Deposition on Soil GHG Fluxes

In this study, we found that the $CH_4$ uptake capabilities of different types of alpine grasslands varied greatly with different N deposition levels. The uptake of $CH_4$ in cultivated grasslands was much greater than that in natural grasslands, i.e., alpine grasslands and alpine meadows. Similarly, Guo et al. (2016) found that the uptake of $CH_4$ fluxes in the cultivated grasslands was 31.3% higher than that in the alpine meadows on the

Qinghai-Tibetan Plateau, though the difference between the two types of grasslands was not significant. This phenomenon can be explained by the fact that $CH_4$ maintains a higher oxidation rate, which may be due to the rich organic carbon in the alpine meadow soil on the Qinghai-Tibetan plateau as well as the high methanogenic content [37–39]. The oxidation of soil-derived $CH_4$ is carried out by oxidizing bacteria, which are mostly specialized aerobic bacteria and are mesophilic microorganisms with a suitable temperature of 20–30 °C. The oxidation of $CH_4$ associated with the survival of methanogenic and methane-oxidizing bacteria may vary with climate, soil, and vegetation [40,41].

In this study, we found that the N addition gradient showed no significant effect on $CO_2$ emissions in the alpine steppe (AS), but soil pH was significantly and positively correlated with $CO_2$ emissions. This can be explained by the fact that N application significantly increased effective N and microbial biomass N, but decreased soil pH (acidifying the soil), resulting in the inhibited root respiration and reducing $CO_2$ emissions [42]. In this study, we found a significant positive correlation between soil phosphorus content and $CO_2$ emissions in the cultivated grassland (CG), which is inconsistent with the findings of Wright et al. (2001). It is possible that the increased soil microbial heterotrophic and autotrophic respiration of soil P under N deposition led to greater soil $CO_2$ emissions [43]. Most importantly, we found that the cultivated grassland produced more $CO_2$ than the natural grasslands—the alpine meadow and the alpine steppe—implying that grassland cultivation may contribute to $CO_2$ emission on the Qinghai-Tibetan Plateau in the era of increasing N deposition.

In this study, we found that $N_2O$ fluxes were significantly influenced by the grassland type, i.e., the cultivated grassland (CG) produced higher $N_2O$ than the natural grasslands of the alpine meadow and the alpine steppe; this is consistent with the findings of Guo et al. (2016) [44]. The possible reason may be that the CG produces more litter than the natural grasslands [45]. In addition, we found a significant positive correlation between soil $NO_3^-$-N content and $N_2O$ fluxes, which is consistent with previous findings [46].

## 5. Conclusions

In this study, we found that the alpine grassland ecosystem on the Tibetan Plateau continuously absorbed $CH_4$ and emitted $CO_2$ and $N_2O$ throughout the peak growing season, which means that the alpine grassland ecosystems act as sinks for $CH_4$ and as sources of $CO_2$ and $N_2O$. Moreover, we found that simulated N deposition had no significant effect on the GHG emissions from the soils of different alpine grasslands, while grassland type had an extremely significant effect on GHG emissions. The uptake of $CH_4$ and the emissions of $CO_2$ and $N_2O$ were greater in the cultivated grasslands than in the natural grasslands of alpine meadow and alpine steppe. GHG fluxes in the alpine grassland soils were closely related to some key soil factors, including soil pH, soil P, and soil$NO_3^-$-N content. Our study has implications on land-use management on the QTP to cope with climate change; i.e., grassland cultivation should be controlled in order to reduce GHG emissions on the QTP in the era of increasing N deposition. Longer-term field observations are still needed in the future to re-testify such findings at a time scale.

**Author Contributions:** Investigation, J.X., Z.M., R.Z., X.H., Z.W. and H.Z.; methodology, H.Z.; visualization, H.S. (Hang Shi); writing—original draft, H.S. (Hang Shi); writing—review and editing, H.S. (Hao Shen) and S.D. All authors have read and agreed to the published version of the manuscript.

**Funding:** This research was funded by the Innovation Platform Construction Project of Qinghai Province of China, grant number [2020-ZJ-T07], the National Key R&D Program of China, grant number [2021FED1124000], the Second Tibetan Plateau Scientific Expedition and Research Program, grant number [2019QZKK0307], and the National Science Foundation of China, grant number [U20A2007-01].

**Data Availability Statement:** The data presented in this study are available on request from the corresponding author. The data are not publicly available due to partial contents of an ongoing research project.

**Conflicts of Interest:** The authors declare no conflicts of interest. The funder had no role in the design of the study; in the collection, analyses, or interpretation of data; in the writing of the manuscript, or in the decision to publish the results.

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
