# Peer review of "Six Years of Grassland Cultivation Promotes CO2, N2O Emissions and CH4 Uptake with Increasing N Deposition on Qinghai-Tibetan Plateau"

_sustainability, doi:10.3390/su141811434_

Round 1

Reviewer 1 Report

In this manuscript, Shi et al. investigated the fluxes of GHGs in three types of alpine grassland by control experiments.The experiment conducted appropriately and obtained results were almost reasonable. However, (1) generally, fluxes of GHGs regulated by soil moisture and temperature, moreover, anthors should give us more information about study sites, such as altitude etc. what's the difference between these three site? why did authors choose these three site? (2) It is hard to say the sink or sources for three type of GHGs, because of the results just showed us the fluxes, not the emission of a period of time. Moreover, authors did not tell us the sample time of day.

Author Response

Dear Editors and Reviewers:

We sincerely thank the editor and all reviewers for their valuable feedback. And we have carefully revised our manuscript entitled “Six years of Grassland cultivation promotes CO2, N2O emissions and CH4 uptake with increasing N deposition on Qinghai-Tibetan Plateau” (ID: sustainability-1824479). Reviewers’ comments are all valuable and very helpful for revising and improving our paper. The reviewers’ comments are laid out below in italicized font and specific concerns have been numbered. Our responses are given in normal font and the “Track Changes” function is used for changes/ supplements to the manuscript. The responses to reviewer’s comments are listed as follows:

Responses to Reviewer #1

1、Generally, fluxes of GHGs regulated by soil moisture and temperature, moreover, anthors should give us more information about study sites, such as altitude etc. what's the difference between these three site? why did authors choose these three site?

Response

The details of the study site, such as longitude and latitude, are indicated in lines 87 and 88 of the text, and the altitude is indicated in lines 88 and 92. In addition, we added the annual precipitation and evaporation information of the two places in line 89-90 and line 93-94.

This study selected three grassland types: alpine grassland, alpine meadow and artificial grassland. Among them, the alpine meadow is located in Xihai Town, Haiyan County, Qinghai Province (100°95'E, 36°93'N), and the grassland type there is a typical alpine meadow with clay based soil. The dominant species is Kobresia capillifolia (Decne.) C. B. Clarke.  The region has a continental climate, with an annual average temperature of 1.4℃, an annual precipitation of 330-370mm, and an annual evaporation of 1400mm (Shen et al., 2009). The alpine grassland is located in tiebujia Town, Gonghe County, Qinghai Province (99°35'E, 37°05'N). the annual average temperature in tiebujia Town is about -0.4-1.2℃, the annual average rainfall is about 360- 430 mm, and the annual evaporation is about 1550 mm. The typical vegetation here is alpine grassland, mainly Stipa and Kentucky bluegrass. The soil is mostly loam clay (Li et al.,2022). The cultivated grassland we selected is also located in Tiebujia Town, and the climate and soil conditions are the same as those of alpine grassland, but we planted Elymus nutans in this place in 2012. Different greenhouse gas emissions may be related to different factors, such as climate, soil texture, vegetation type, etc. (Sommer et al.,2004; Wei et al.,2014). Because the above three grassland types are the main grassland types in the Qinghai Tibet Plateau, the study of greenhouse gases on different grassland types will help to better understand the mechanism of greenhouse gas emissions from grassland in the Qinghai-Tibetan Plateau.

Related references:

Shen H, Dong S, DiTommaso A, et al. Eco-physiological processes are more sensitive to simulated N deposition in leguminous forbs than non-leguminous forbs in an alpine meadow of the Qinghai-Tibetan Plateau. Science of The Total Environment, 2020, 744: 140612.

Li S, Dong S, Shen H, et al. Synchronous responses of plant functional traits to nitrogen deposition from dominant species to functional groups and whole communities in alpine grasslands on the Qinghai-Tibetan Plateau. Frontiers in plant science, 2022: 110.

Sommer, S.G.; Schjoerring, J.K.; Denmead, O.J.A.i.a. Ammonia emission from mineral fertilizers and fertilized crops. Advances in agronomy 2004, 82, 82008-82004.

Wei, D.; Liu, Y.; Wang, Y.; Wang, Y.J.G. Three-year study of CO2 efflux and CH4/N2O fluxes at an alpine steppe site on the central Tibetan Plateau and their responses to simulated N deposition. Geoderma 2014, 232, 88-96.

2、It is hard to say the sink or sources for three type of GHGs, because of the results just showed us the fluxes, not the emission of a period of time. Moreover, authors did not tell us the sample time of day.

Response

Thanks for your kind and constructive suggestions.

Firstly, though the results just showed the fluxes, it has an implication for envaluating whether it is sink or source in the future. And this can provide reference for future GHGs study in the alpine grassland ecosystem.

Second, the Methods is scientific and many previous studies can support this method in GHGs study.

In detail, our greenhouse gases were collected in August, the peak of the 2020 growth season. Sampling was carried out on three consecutive sunny days from 9 a.m. to 11 a.m. Use an opaque PVC static box with a diameter of 20 cm and a height of 25 cm for sampling. The gas fluxes were collected in the chamber with a 50ml plastic syringe at 0, 5, 10, 15, 20 and 25min after closing the static chamber. The collected gas samples are sealed in aluminum foil sampling bags and transported to the laboratory within 24 hours. This study refers to the methods of Zhang et al., 2013、Li et al., 2014 and Zhao et al., 2017.

Finally, the source and sink issues mentioned in this article have similar statements in other articles, such as "In the whole growing season, the average CH4 absorption of alpine meadow soil in natural state is (35.40±1.92) μg*m-2h-1, indicating that the alpine meadow soil is the net absorption sink of atmospheric CH4 " Zhang et al., 2013;"The positive values of CO2 fluxes in all the experiments of the present study indicated that the alpine grasslands were the source of CO2 ."Zhao et al., 2017; "Also, the differences in soil pH can also affect CH4 fluxes through controlling activity of methanogens and methanotrophs, and inhibit the CH4 oxidation. CH4 emissions generally occurred during the early growing season of alpine plants (May) and the non-growing season of alpine plants (October), while CH4 sink happened in the peak growing season of alpine plants (August)"Li et al., 2014.

In addition, we have added the sampling time in line 113 of the text.

Related references:

Zhang, P.; Fang, H.; Cheng, S.; Xu, M.; Li, L.; Dang, X.J.A.E.S. The early effects of nitrogen addition on CH4uptake in an alpine meadow soil on the Eastern Qinghai-Tibetan Plateau. Acta Ecologica Sinica 2013, 33, 4101-4110.

Li, Y.; Dong, S.; Liu, S.; Zhou, H.; Gao, Q.; Cao, G.; Wang, X.; Su, X.; Zhang, Y.; Tang, L.; et al. Seasonal changes of CO2, CH4 and N2O fluxes in different types of alpine grassland in the Qinghai-Tibetan Plateau of China. Soil Biology and Biochemistry 2015, 80, 306-314.

Zhao, Z.; Dong, S.; Jiang, X.; Liu, S.; Ji, H.; Li, Y.; Han, Y.; Sha, W. Effects of warming and nitrogen deposition on CH4, CO2 and N2O emissions in alpine grassland ecosystems of the Qinghai-Tibetan Plateau. Sci Total Environ 2017, 592, 565-572

Reviewer 2 Report

Comments and suggestions for Authors

Six years of Grassland cultivation promotes CO2, N2O emissions and CH4
uptake with increasing N deposition on Qinghai-Tibetan Plateau

Subject is interesting and fall within the scope of the journal. The experimental dataset undoubtedly are useful and constitutes scientific values.

In this study investigated how N deposition affected GHGs fluxes in the alpine meadow, alpine steppe, and cultivated grassland during the growing season. The manuscript requires minor additions and corrections.

General remarks

In order to increase the usefulness of the article, Authors must refer to the following points.

      Abstract - Line 23 and 24 – (P > 0.05 or (P < 0.05) ???

      Keywors - I propose to verify the keywords (repetitions from the title of the manuscript).

      Materials and Methods - Atmospheric data (precipitation and temperature) in months and years of research should be supplemented. This is very important in the context of the fluxes of GHGs. Line 100 - make up % N in ammonium nitrate. Lines 101-102 and 103 - verify the record of units. Line 109 - indicate the day, month and year of sampling. Subsection 2.4. - complete the soil sampling date. It is very important in determining the available forms of the elements.

      Results – Line 141 and 143 (P > 0.05 or (P < 0.05) ??? Table 1 – ET what is it - to explain.

      Discussion - The discussion of the results is generally well written. The only problem is the varied results obtained by other authors, conducted in the same climatic and soil conditions. Maybe it's the effect of a different nitrogen fertilization? or meteorological conditions (temperature, precipitation) in the years of the research? It should be clarified.

      Conclusion – Line 278-280 It is not possible to write about the entire growing season, when samples were taken after six years of research for three consecutive days in the peak of the growing season. The applied small doses of N did not allow to determine the influence of this factor on the fluxes of GHGs.

      References please save correctly in accordance with editorial requirements. The order of References should be noted as cited in the manuscript text.Publisher names are missing from some References.

You need to suplement: Author Contributions and Conflicts of Interest

Specific comments

Line 111 and 118 – it should be Figure 1.

Line 124  and 125 – Kalium…Sulfur ????

Line 127 - this is not the place for References [26].

Line 145 and 147 - it should be (Fig. 2a), (Fig. 2b)

Line 148 – DNL or NDL???

Line 152 - it should be Figure 2.

Line 155, 169, 182 - units must be checked.

Line 160, 173 -  (P > 0.05)???

Line 162 and 164 - it should be Figure 3a and Figure 3b.

Line 166 - it should be Figure 3.

Line 175 and 177 - it should be Figure 4a and Figure 4b.

179 - it should be Figure 4.

Line 186, 188 and 190 - it should be Figure 5a,  Figure 5b and Figure 5c.

Line 193 - it should be Figure 5.

Line 196 and 200 - (Sulfur)????

Table 2 - P  ..-.614* ????

Line 210 - it should be (Fig 2a).

The entire References  must be adapted to the publishing requirements.

Manuscript contains interesting results, but requires improvement before publication.

Best regards

Author Response

Dear Editors and Reviewers:

We sincerely thank the editor and all reviewers for their valuable feedback. And we have carefully revised our manuscript entitled “Six years of Grassland cultivation promotes CO2, N2O emissions and CH4 uptake with increasing N deposition on Qinghai-Tibetan Plateau” (ID: sustainability-1824479). Reviewers’ comments are all valuable and very helpful for revising and improving our paper. The reviewers’ comments are laid out below in italicized font and specific concerns have been numbered. Our responses are given in normal font and the “Track Changes” function is used for changes/ supplements to the manuscript. The responses to reviewer’s comments are listed as follows:

Responses to Reviewer #2

1、Abstract- Line 23 and 24 – (P > 0.05 or (P < 0.05) ???

Response:Sorry, this is a mistake, we have corrected it in line 23 and line 24.

2、Keywords -I propose to verify the keywords (repetitions from the title of the manuscript). 

Response:According to your suggestion, we have modified it in line 31.

3、Materials and Methods- Atmospheric data (precipitation and temperature) in months and years of research should be supplemented. This is very important in the context of the fluxes of GHGs. Line 100 - make up % N in ammonium nitrate. Lines 101-102 and 103 - verify the record of units. Line 109 - indicate the day, month and year of sampling. Subsection 2.4. - complete the soil sampling date. It is very important in determining the available forms of the elements.

Response:According to your opinion, we supplemented the atmospheric temperature, annual precipitation and annual evaporation of the region in line 89-90 and line 93-94. And we added the nitrogen content of ammonium nitrate in line 103. We verified and changed the units of lines 106. And we added the time of gas sampling (line 111-113) and soil sampling (line 123).

4、ResultsLine 141 and 143 (P > 0.05 or (P < 0.05) ??? Table 1 – ET what is it - to explain.

Response:According to your suggestion, we have modified it (line 142-144). ET is our mistake and has been corrected (Table 1).

5、Discussion -The discussion of the results is generally well written. The only problem is the varied results obtained by other authors, conducted in the same climatic and soil conditions. Maybe it's the effect of a different nitrogen fertilization? or meteorological conditions (temperature, precipitation) in the years of the research? It should be clarified.

Response:Thank you for your valuable suggestions. We followed your suggestions and discussed whether the results were inconsistent due to different nitrogen fertilizers or different meteorological conditions. We added this part to lines 221-222 and 246-248.

6、ConclusionLine 278-280 It is not possible to write about the entire growing season, when samples were taken after six years of research for three consecutive days in the peak of the growing season. The applied small doses of N did not allow to determine the influence of this factor on the fluxes of GHGs.

Response:Yes, we followed your suggestion and made changes on line 285 of the text. On the second question, I agree very much that it is possible that our nitrogen application rate is too small. Our nitrogen application rate is the same as that of Zhu etal., and she also reached similar conclusions.

Related references:

Zhu, Xiaoxue; Luo, Caiyun; Wang, Shiping. Effects of warming, grazing/cutting and nitrogen fertilization on greenhouse gas fluxes during growing seasons in an alpine meadow on the Tibetan Plateau. Agricultural and Forest Meteorology 2015, 214-215, 506–514.

7、References-please save correctly in accordance with editorial requirements. The order of References should be noted as cited in the manuscript text.Publisher names are missing from some References.

Response:Sorry, it has been changed according to your comments and editing requirements.

8、You need to suplement: Author Contributions and Conflicts of Interes

Response:We followed your advice and added the Author Contributions (lines 297–299) and Conflicts of Interest (lines 303–305) in the text

9、Line 111 and 118 – it should be Figure 1. 

Response:Yes, we have changed this content on line 114 and 121.

10、Line 124 and 125 – Kalium…Sulfur ????

Response:We are very sorry for our incorrect writing, and we have made changes on line 127.

11、Line 127 - this is not the place for References [26].

Response:It has been modified on line 129 according to your suggestion.

12、Line 145 and 147 - it should be (Fig. 2a), (Fig. 2b)

Response:Yes, we have made changes in line 146 and line 148 according to your comments.

13、Line 148 – DNL or NDL???

Response:Here is N disposition level, which should be NDL, and has been modified in line 149 and Table 1.

14、Line 152 - it should be Figure 2.

Response:It has been modified on line 153 according to your suggestion.

15、Line 155, 169, 182 - units must be checked.

Response:Sorry, we checked the unit and modified it in line 156-157, line 171 and line 184-185.

16、Line 160, 173 - (P > 0.05)???

Response:Sorry, this is a mistake, we have corrected it in line 159 and line 162.

17、Line 162 and 164 - it should be Figure 3a and Figure 3b.

Response:Yes, it has been corrected in line 164 and 166 according to your comments.

18、Line 166 - it should be Figure 3.

Response:It has been modified on line 168 according to your suggestion.

19、Line 175 and 177 - it should be Figure 4a and Figure 4b.

Response:Yes, we made corrections on lines 177 and 179 of the text.

20179 - it should be Figure 4.

Response:Yes, we corrected it on line 181 of the text.

21、Line 186, 188 and 190 - it should be Figure 5a, Figure 5b and Figure 5c.

Response:we have made changes in line 189、191 and 193 according to your comments.

22、Line 193 - it should be Figure 5.

Response:Yes, we corrected it on line 195 of the text.

23、Line 196 and 200 - (Sulfur)????

Response:This part is the content of soil sulfur. We refer to the literature of Zhao et al.(2022).

Related references:

Zhao, W., Xiao, C., Li, M., Xu, L., Li, X., & He, N. Spatial variation and allocation of sulfur among major plant organs in China. Science of The Total Environment 2022, 157155.

24、Table 2 - P  ..-.614* ????

Response:Sorry, this is a mistake, we have corrected it in Table 2.

25、Line 210 - it should be (Fig 2a).

Response:Yes, we corrected it on line 212-213 of the text.

Round 2

Reviewer 1 Report

All of my concerns have been addressed,  I think that it could be recommened for publishing.